# SAFHE: Defending Against Backdoor and Gradient Inversion Attacks in Federated Learning

## Abstract

Federated learning (FL) is an increasingly popular approach in machine learning that enables a set of clients to jointly train a global model without ever sharing their private data, using a central server to aggregate clients' local weight updates. However, previous work has shown that the distributed nature of federated learning makes it susceptible to two major attacks: *backdoor attacks*, where malicious clients submit large weights that incorrectly change model behavior, and *gradient inversion attacks*, where a malicious eavesdropper is able to reconstruct the clients' training data by viewing the weight updates sent by clients to the central server. Although various solutions have been proposed in the literature that defend against these two attacks separately, present approaches remain largely incompatible, creating a trade-off between defending against the two types of attacks. This poses a major challenge in deploying FL in privacy-sensitive ML applications.

We present SAFHE (Secure Aggregation with Fully Homomorphic Encryption), a novel scheme to defend against *both* backdoor attacks and gradient inversion attacks. Our secure aggregation method combines the use of fully homomorphic encryption (FHE) and the gradient norm clipping defense to defend against large malicious client updates, by pre-weighting client updates using a function that can be evaluated in the encrypted domain. This allows the server to reject large-magnitude updates without seeing their cleartext values. We demonstrate that Chebyshev approximations of a product of sigmoids work for this purpose, and perform simulations suggesting that such a scheme can defend against backdoor attacks without significantly impacting model accuracy. Additionally, we show that these approximations can be accurately and efficiently computed in the encrypted domain.

## 1 Introduction

Federated learning (FL) makes it possible to train Machine Learning (ML) models in a distributed fashion, while keeping data local to client devices Konečný et al. (2016); McMahan et al. (2017). In an FL framework, a central server and a set of clients jointly train a global model. In every round, the clients train locally with their private data and then submit an updated model to the central sever, which averages the updates into the new joint model Bagdasaryan et al. (2020).

However, recent work has shown that FL is susceptible to a range of privacy attacks, thus posing a major barrier into the adoption of FL systems in ML applications. Though the data stays local to client devices, the scheme is still vulnerable to certain types of attacks. Notably, an attacker who can see weight updates sent by clients to a central server can use *gradient inversion* attacks to reconstruct the clients' training data Zhu et al. (2019); Geiping et al. (2020); Yin et al. (2021a); Huang et al. (2021); Yin et al. (2021b). One way to contend with this problem is to use a fully homomorphic encryption (FHE) scheme Gentry (2009); Jin et al. (2023). When using FHE, the clients submit encrypted updates to the central server, which can then aggregate the weight updates from the clients without ever seeing the plain-text updates or the decryption keys. As a result, this system prevents gradient inversion attacks.

However, FHE does not prevent against *backdoor attacks* by malicious clients Bagdasaryan et al. (2020). More specifically, Bagdasarya et al. showed that federated learning is generically vulnerable to model poisoning Bagdasaryan et al. (2020). Indeed, the aggregation step in FL is very sensitive to malformed gradient, and even just one gradient with a very high norm from one client can arbitrarily bias the global model Rathee et al. (2022). Many backdoor attacks leverage large gradients sent by malicious attacks to poison the model Shejwalkar et al. (2022); Baruch et al. (2019); Fang et al. (2020); Shejwalkar & Houmansadr (2021). Defending against backdoor attacks is a notoriously hard problem: no current defenses are powerful enough to completely stop all attacks Rathee et al. (2022); Shejwalkar et al. (2022). Recent work on backdoor attacks has shown that filtering gradients based on their $\ell_2$ norm when aggregating (which we call the *gradient norm clipping defense* in this paper), even if simple, is the most effective defense against a large number of practical and sophisticated backdoor attacks Rathee et al. (2022); Shejwalkar et al. (2022); Sun et al. (2019); Bell et al. (2023); Lycklama et al. (2023), even in the case of *adaptive* and *untargeted* attacks Shejwalkar et al. (2022). Therefore, even though there is no panacea for solving backdoor attacks Wang et al. (2020), our work rests on the well-studied claim that sending ill-formed gradients is the most successful backdoor attack in practice, and that the most effective defense is the norm clipping defense.

However, the norm clipping defense proves to be difficult with encrypted weight updates: FHE schemes can only efficiently perform addition and multiplication in the encrypted domain, but they cannot evaluate conditional branches, so large updates cannot be treated differently from normal ones Sun et al. (2018). This seemingly presents an incompatibility between defending against backdoor and gradient inversion attacks. In this paper, we present a new method to reject large weight updates in the FHE domain without conditional branches: instead of simply averaging the updates, the central server first applies a function to each client's updates to compute that update's "weight," where large updates have weights close to 0, and are thus rejected. Finding an appropriate weighting function is nontrivial, as only polynomials can be efficiently evaluated homomorphically. We employ two different approximation techniques—Chebyshev and Minimax—to find suitable functions. We show through simulations that these approximations successfully defend against backdoor attacks without impacting the model accuracy. Additionally, we show that our approximations can be accurately and efficiently homomorphically evaluated.

## 1.1 RELATED WORK

With the increasing popularity of federated learning, there have been many strategies proposed for secure aggregation methods. The main privacy-preserving techniques that have been considered for this purpose include multi-party computation Fereidooni et al. (2021), differential privacy Stevens et al. (2021), Shamir's secret sharing Bonawitz et al. (2016), fully homomorphic encryption Fereidooni et al. (2021), or a combination of these Rathee et al. (2022). All these methods protect against gradient inversion attacks, but they do not consider the threat of malicious clients.

We are not the first to propose a method that attempts to defend against gradient inversion and backdoor attacks simultaneously; we now highlight some of the differences between our SAFHE method and prior work. First, most of the recently proposed methods rely on zero-knowledge proofs (ZKP), general-purpose secure multi-party computation (MPC), and other heavy cryptographic primitives Lycklama et al. (2023); Rathee et al. (2022); Bell et al. (2023); Roy Chowdhury et al. (2022); Bonawitz et al. (2016); Truex et al. (2019); He et al. (2020); So et al. (2020). While FHE also has its limitations and can be inefficient in certain cases, we believe that it is beneficial to diversify the array of cryptographic primitives considered in secure FL, especially given that the efficiency of each of these cryptographic techniques is evolving separately. A major advantage of using our method SAFHE rather than the methods that use ZKP, MPC, or Shamir secret sharing is that we do not have any communication overhead, given that the FHE scheme has optimal communication costs. Using these other cryptographic primitives also makes the problem of clients dropping out during training much harder to fix, given that the protocols usually require all of the parties to be present. This is, however, not concern in our client-server one-shot method. Lastly, another importance difference is that some of these recent methods, such as So et al. (2020), require two non-colluding servers, which is known to be an issue in practice Rathee et al. (2022). SAFHE only requires one server, which corresponds to the standard way of performing FL.

Regarding the efficiency of FHE evaluations, a lot of interest has recently emerged in finding polynomial approximations of activation functions, to use FHE for machine learning models, which is

a technique that we use in our SAFHE method. For this reason, CryptoNets, a Microsoft library for neural networks that can be applied to encrypted data, uses a square function to approximate the sigmoid Gilad-Bachrach et al. (2016), while Ali et al. (2020) use $x^2 + x$ to approximate the ReLU activation function for efficient FHE evaluation. While the work of Khan et al. consider more complex approximations for activation functions by using Chebyshev polynomials instead Khan et al. (2021), we are the first to use this technique for secure weight update aggregation in FL training.

## 1.2 PROBLEM TO SOLVE

Our work contends with a classification model in an FL framework and an IID environment. Our threat model is an attacker attempting to carry out two different types of attacks, which are variations of the sending ill-formed gradients attack discussed in the inttroduction:

1. **Noise attack.** The attacker's goal is to degrade the model's general accuracy. To do so, she submits weight updates consisting of uniformly distributed noise on some large interval. The magnitudes of these weights can be far larger than those present in the model, so they have an outsized impact on the central weights.

2. **Switch classes attack.** The attacker's goal is to influence the model to invert the classifications of two particular classes. For example, the attacker might wish to force an image classification model to misclassify airplanes as birds and vice versa. Notably, the attacker does *not* wish to influence the model's behavior on data in other classes, so that the attack may go unnoticed by other users. This attack is representative of real-world attacks on image classification models Gong et al. (2022).

Each of these attacks may be carried out in a *one shot* manner, where the attacker sends one malicious update, or *continuously*, where the attacker persists in sending malicious updates for a period spanning multiple rounds. To carry out both of these attacks, attackers rely on being able to submit updates whose difference from the current global model is large in magnitude.

## 2 OUR PROPOSED APPROACH: SAFHE

In this paper, we present SAFHE: Secure Aggregation with Fully Homomorphic Encryption, a novel scheme to defend against *both* backdoor attacks and gradient inversion attacks. Our strategy involves computing the appropriate "weighting" of each client's update by applying some function $H$ before averaging. $H$ should, essentially, return 1 for updates that are safe and 0 for updates that are not. The difficulty lies in finding a function $H$ that can be computed homomorphically.

The high-level idea of our approach is the following. In an FHE environment, we cannot see the cleartext weight updates but we can evaluate functions (homomorphically) on them. That is, we can prevent large model updates without seeing the cleartext weights. Low-degree polynomials can be efficiently evaluated in an FHE environment while maintaining high accuracy, so FHE coupled with our secure aggregation method defends against both gradient inversion and backdoor attacks.

## 2.1 NOTATION AND ROADMAP

In this section, we provide a high-level description of the SAFHE method, assuming we have chosen an appropriate weighting function $H : \mathbb{R} \to [0, 1]$. Later sections will discuss how we chose an $H$ that is both secure against large updates (Section 2.2) and efficient to compute through FHE (Section 2.3). We note that while in this section we write $w_i \in \mathbb{R}$ for simplicity in the presentation, our $H$ function is applied to the $\ell_2$ norm of the client's vector of weights.

Assume that $(E, D, EVAL)$ is a FHE-scheme, where given ciphertexts $E(x_1), E(x_2), ..., E(x_k)$ and a function $f$, $EVAL$ can fully homomorphically evaluate $f$, i.e. $EVAL(f, E(x_1), E(x_2), ..., E(x_k)) = E(f(x_1, ..., x_k))$. Moreover, assume $H : \mathbb{R} \to [0, 1]$ is a function that can be efficiently evaluated through FHE. That is, given the encryption $E(x)$ of any plaintext $x$, the central server is able to efficiently compute $EVAL(H, E(x)) = E(H(x))$. We denote that the set of clients as $\mathcal{C} = [|\mathcal{C}|]$ and the update sent by client $i \in \mathcal{C}$ as $w_i$. In that case, define a (non-secure) averaging of these updates as $\nabla(w_1, ..., w_{|\mathcal{C}|}) = \frac{1}{|\mathcal{C}|} \sum_{i \in \mathcal{C}} w_i$. Instead,

we would like to calculate the secure aggregation $\nabla_{sec}(w_1, ..., w_{|\mathcal{C}|}) = \frac{1}{|\mathcal{C}|} \sum_{i \in \mathcal{C}} H(w_i) \cdot w_i$, assuming $H$ successfully rules out any malicious updates by mapping them to 0. Lastly, define $mult : \mathbb{R} \times \mathbb{R} \to \mathbb{R}$ as the multiplication function: $mult(x_1, x_2) = x_1 \cdot x_2$. Then, in each round of training, SAFHE computes this average as described in Algorithm 1.

---

**Algorithm 1** Secure Aggregation with Fully Homomorphic Encryption (SAFHE)

---

**input:** Encrypted weight updates $\{E(w_i)\}_{i \in \mathcal{C}}$, a weighing function $H : \mathbb{R} \to (0, 1)$
**output:** Encryption of the secure aggregation $E(\nabla_{sec}(\{w_i\}_{i \in \mathcal{C}}))$
1: **procedure** SAFHE($\{E(w_i)\}_{i \in \mathcal{C}}$)
2:     **for** $i \in \mathcal{C}$ **do**
3:         $x_i \leftarrow EVAL(H, E(w_i))$                               $\triangleright$ Compute $E(H(w_i))$
4:         $y_i \leftarrow EVAL(mult, x_i, E(w_i))$                 $\triangleright$ Compute $E(H(w_i) \cdot w_i)$
5:     **end for**
6:     $z \leftarrow EVAL(\nabla, \{y_i\}_{i \in \mathcal{C}})$.                 $\triangleright$ Compute $E(\nabla_{sec}(w_1, ..., w_{|C|}))$
7:     **return** $z$.
8: **end procedure**

---

Note that line 3 of Algorithm 1 can be computed efficiently by assumption that $H$ can be efficiently evaluated through FHE. As such, each $x_i$ gets assigned $E(H(w_i))$. Since multiplication can be evaluated efficently with FHE, line 4 correctly and efficently computes $EVAL(mult, x_i, E(w_i)) = EVAL(mult, E(H(w_i)), E(w_i)) = E(mult(H(w_i), w_i)) = E(H(w_i) \cdot w_i)$, which gets assigned to $y_i$. Since the non-secure averaging function $\nabla$ simply involves a sum of variables and a product by a constant ($1/|\mathcal{C}|$), it can also be evaluated efficiently via FHE, so line 6 correctly computes:

$$EVAL(\nabla, \{y_i\}_{i \in \mathcal{C}}) = EVAL(\nabla, \{E(H(w_i) \cdot w_i)\}_{i \in \mathcal{C}}) = E(\nabla(\{H(w_i) \cdot w_i\}_{i \in \mathcal{C}})) \tag{2.1}$$

$$= E\left(\frac{1}{|\mathcal{C}|} \sum_{i \in \mathcal{C}} H(w_i) \cdot w_i\right) = E(\nabla_{sec}(\{w_i\}_{i \in \mathcal{C}})) \tag{2.2}$$

which gets assigned to $z$. As such, our algorithm runs efficently and correctly returns $E(\nabla_{sec}(\{w_i\}_{i \in \mathcal{C}}))$, the encryption of the secure aggregation. Notice that throughout the algorithm, the central server running the protocol never gets access to the plaintext updates $w_i$, ensuring the privacy of the clients. Assuming that the decryption $D$ is available to the clients, they can then incorporate the encrypted global model update they receive into their private models.

## 2.2 IDEAL THRESHOLD FUNCTION

Ideally, $H$ would be a square pulse function, which we can represent as a product of two sigmoids:

$$H_{a,b,c}(x) = \frac{1}{1 + e^{-(x-a)/c}} \cdot \frac{1}{1 + e^{-(x-b)/c}}. \tag{2.3}$$

We define $H$ to be a product of sigmoids because this is the simplest function that we found that is *continuous*, which is a property that we require in order to be able to use FHE. The $a$ parameter determines the left threshold, the $b$ parameter determines the right threshold, and the $c$ parameter determines how steep the transition between 0 and 1 are (see Figure 2 for an example).[1] The crucial point is that applying the function $H$ to the client servers is equivalent to the gradient norm clipping defense (minus a small error around $a$ and $b$), and therefore the claims made in the literature about the effectiveness of this defense carry on to our setting, which implies that SAFHE does defend against both gradient inversion attacks and backdoor attacks in theory. Hence, the empirical evaluation is primarily concerned with finding an appropriate trade-off between the degree of the polynomial and the efficiency of the FHE evaluation.

---

[1]Regarding the choice of the $a, b$ parameters, the right parameters are different in each setting and architecture. We emphasize that choosing the right width of $H$ is a problem concerned with the gradient norm clipping defense from the backdoor attacks literature, rather than a bug of our SAFHE method, and hence we defer to the literature for making this choice. In a similar fashion, the ELSA method states that selecting an appropriate gradient bound is orthogonal to the problem of making the method compatible with gradient inversion defenses Rathee et al. (2022). A proposed way for choosing the appropriate gradient norm bounds from the literature is to use a variant of the median of the medians method to select the bounds Lycklama et al. (2023).

## 2.3 FINDING AN $H$ WITH EFFICIENT FHE EVALUATION

While the $H$ defined above rejects large updates as desired, it, itself, cannot be evaluated efficiently in current FHE systems. As a result, we need to find a function $H$ in terms of only addition and multiplication operations—in other words, $H$ must be a polynomial. Otherwise, the function $H$ as defined above requires division and exponentiation. For $H$ to be efficient to evaluate, it must be approximated as a polynomial. In theory, this approximation is guaranteed to be possible: by the Stone-Weierstrass theorem, any continuous function on a closed interval can be uniformly approximated by polynomials De Branges (1959). However, Stone-Weierstrass alone does not give us a way to actually find these approximations. To find these approximations, we employ two methods from numerical analysis literature: Chebyshev polynomials and Minimax polynomials. We favor these methods above others (e.g., Taylor approximations) because these methods allow us to approximate $H$ over an arbitrarily large interval $[a, b]$, the entire domain of our FHE environment.

## 2.4 THE CHEBYSHEV AND MINIMAX APPROXIMATIONS

### 2.4.1 CHEBYSHEV POLYNOMIALS

The Chebyshev polynomials of the first kind are defined recursively as

$$T_{n+1}(x) = 2xT_n(x) - T_{n-1}(x), \quad n \geq 1,$$

with base cases $T_0(x) = 1$ and $T_1(x) = x$. The Chebyshev approximation theorem states the following Press et al. (2007): Let $f(x)$ be an arbitrary function in the interval $[-1, 1]$ and let the $N$ Chebyshev coefficients $c_j$ be defined as

$$c_j = \frac{2}{N} \sum_{k=1}^{N} f(x_k) T_{j-1}(x_k).$$

Then, we can approximate the function $f$ as

$$f(x) \approx \Big[ \sum_{k=1}^{N} c_k T_{k-1}(x) \Big] - \frac{1}{2} c_1.$$

This approach can be generalized to an arbitrary range $[a, b]$ instead of $[-1, 1]$ by scaling the input to the function. Given that we find Chebyshev polynomials more appropriate to use in practice, we defer the explanation on minimax polynomials to the full version of the paper.

## 2.5 SIMULATION

To successfully evaluate our SAFHE method, we need to perform to following analysis:

1. How well do the Chebyshev and Minimax approximations approximate the $H$ function? What degree of polynomial is needed to make this approximation sufficiently accurate?

2. How much time is required and how much error is introduced when evaluating our approximations homomorphically?

3. In practice, how well do these defenses work? That is, are they robust to our threat model? How do they influence training rate and accuracy compared to a simple averaging aggregation scheme?

While item (2) above requires a FHE implementation of our aggregation functions, the rest of these measurements can be made in a *simulated* FL environment, without actually using any encrypted data. We implement a federated learning simulation which allows us to experiment with different aggregation functions and approximations, without having to contend with the additional overhead and complexity introduced by fully homomorphic encryption. Since in this simulation we use aggregation functions that we verify can be accurately and efficiently evaluated homomorphically, we know that these results are representative of those performed in an FHE environment.

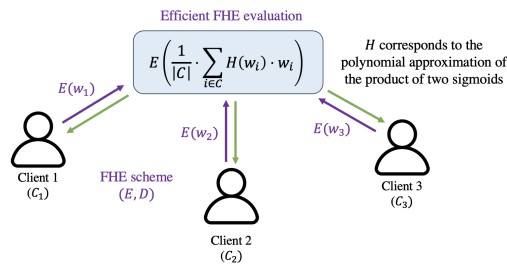

Figure 1: Summary of our SAFHE method for secure aggregation in FL.

## 2.6 SUMMARY OF OUR SAFHE METHOD

In sum, our method SAFHE is summarized in Algorithm 1, where the weighting function $H$ corresponds to the product of two sigmoids (see Equation 2.3). We use the Chebyshev and Minimax polynomial approximations from the numerical analysis literature as described in Section 2.4 to approximate our weighting function $H$, so that the central server can homorphically aggregate the clients' weights in a computationally efficient manner. The SAFHE protocol, which allows the server to identify and reject updates with too-large magnitudes without accessing their cleartext values, is summarized in Figure 1.

We emphasize that the core idea behind SAFHE goes beyond the gradient norm clipping defense and we believe it is widely applicable: If a successful backdoor defense can be captured by a continuous function, then it can be coupled with FHE and Chebyshev or minimax polynomial approximations to simultaneously make it a successful defense against gradient-inversion attacks. For example, we hypothesize that our work might be applicable to split computing, given that the FHE approach can offload work to a server with some privacy preserving properties Dong et al. (2022).

Given the nature of polynomials, it is important to consider how the encryption space compares to the polynomial approximation interval and to the $H$'s function $[a, b]$ interval. For the Chebyshev and minimax polynomials, in order to maximize accuracy, the approximation should be performed over the interval $[a - \epsilon, b + \epsilon]$ for some small value of $\epsilon$. Given that FHE operates using modular arithmetic, one determines an encryption space $[c, d]$ a priori. If there is a large gap between $a$ and $c$, and likewise between $b$ and $d$, then SAFHE is not a reliable method, given that the Chebyshev and minimax polynomials will no longer be close to 0 at the points $x$ such that $x \ll a$ or $x \gg b$. However, we do not believe that this is an issue in practice, given that we can choose the encryption space to be close to $[a, b]$ from the beginning. If the gradient values need to be made smaller, then a possible avenue for future work would be to combine our SAFHE method with quantization techniques Hubara et al. (2017); Reisizadeh et al. (2020).

## 3 EXPERIMENTS PERFORMED

### 3.1 POLYNOMIAL APPROXIMATION

We implemented a library to compute the coefficients of the Chebyshev and Minimax polynomials Python and used it to compute approximations of the weighting function $H_{a,b,c}$ for arbitrary $a, b, c$, and degree of approximation, on any interval of our choice. We observe that degree 10 polynomials already achieve highly accurate approximations (Figure 2). Both our FL simulation and our FHE experiments leverage this library.

### 3.2 EVALUATING POLYNOMIALS IN AN FHE ENVIRONMENT

We evaluate polynomials on encrypted real numbers using the Microsoft SEAL SEAL FHE library, which provides an implementation of the CKKS Cheon et al. (2017) scheme. CKKS supports additions and multiplications, but yields only approximate results with a predetermined precision. It uses a rescaling procedure to stabilize the scale expansion after multiplications, which truncates a ciphertext into a smaller modulus, which then leads to rounding of plaintext. The polynomial

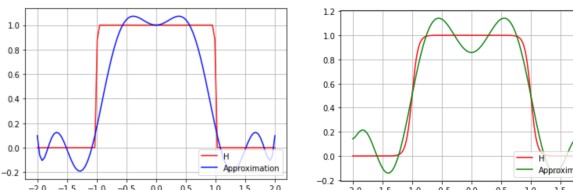

Figure 2: Degree 10 approximation of $H_{-1,1,0.005}$ (in red) over $[-5, 5]$ with Chebyshev (left, blue) and Minimax (right, green) polynomials. Mistake to fix: the red curves should be the same ones but they are not (it is confusing).

modulus degree—the degree of a power-of-two cyclotomic polynomial usually between $2^{10}$ and $2^{15}$—controls the number of rescalings that can be performed. A larger polynomial modulus degree allows for greater multiplicative depth in the computation, enabling more complex encrypted computations and more accurate results, at the cost of runtime. We use the EVA Dathathri et al. (2020) compiler to optimize the FHE computations and select appropriate encryption parameters for SEAL. Regarding the width of H, we choose it empirically by running our set-up first with un-encrypted gradients and determining the "typical" norm of a benign gradient.

### 3.3 Simulation Details

We implemented our FL simulation in Python using Torch. The simulation allows specifying the aggregation strategy as well as the parameters of an attack. Our simulation supports four aggregation schemes: a basic average, the "ideal" rejection scheme using the exact $H$ function, and the Chebyshev and Minimax approximations. In all cases, the ultimate weighting applied to a device's update is determined by the product of the evaluation of the aggregation function on each of its constituent parameters.[2] We empirically determined the size of the largest allowable weight updates on a per-layer basis by performing several rounds of training on the model and setting the bounds to be the largest updates encountered in ordinary training.

We implemented both the "noise" and "switch-classes" attacks outlined in Section 1.2. The "switch-classes" attack uses the following algorithm: the attacker generates a relabeled version of its subset of training data, inverting the classifications for all examples in a pair of classes; it then removes 80% of the examples that belong to other classes, so that the training data primarily consists of members of these two classes. The attacker then trains its local model for 100 epochs on the misclassified data, starting from the model distributed by the central server.

All of our experiments were performed using a 10-layer convolutional neural network performing classification on the CIFAR-10 dataset. We used ReLU as our activation function, cross entropy loss, the SGD optimizer, and a multi-step LR scheduler with $\gamma = 0.1$.

## 4 Results and discussions

### 4.1 FHE Performance

We found that our polynomial approximations can be accurately and efficiently homomorphically evaluated in the CKKS scheme. We use a polynomial modulus degree of $2^{14}$ to restrict the mean squared error between encrypted and unencrypted computation to below $10^{-6}$. Within this polynomial modulus degree, we can evaluate polynomial approximations of degree 2 to 10 efficiently, with runtimes increasing linearly from 0.07 seconds to 0.40 seconds for $2^{13}$ inputs (see Figure 3). If we were to use much larger degree approximations for $H$, we would need to increase the polynomial modulus degree, which would result in an exponential increase in runtime. However, the degree 6 to 10 approximations, which we have shown can be efficiently evaluated with FHE, are indeed sufficient for defending against backdoor attacks (see Section 5.1).

---

[2]We had to multiply by a final constant to eliminate the systematic under-estimation inherent in the approximations.

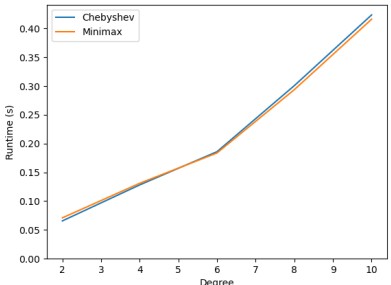

Figure 3: FHE evaluation times (in seconds) of Chebyshev and Minimax polynomial approximations of $H_{-1,1,0.005}$ for even degrees 2 to 10, on 8,192 uniformly-distributed inputs in [-5,5].

## 5 FEDERATED LEARNING TRAINING WITH SAFHE

### 5.1 MODEL ACCURACY AND DEFENSE SUCCESS

We successfully used our simulation to validate our strategy. Ultimately, we found the Chebyshev approximation to be much more effective than the Minimax approximation. Due to the Minimax approximation's goal of bounding maximal error, the resulting approximations end up with large peaks outside of the desired allowable region, which allows well designed updates to slip through. As a result, we present the following results using the Chebyshev approximations. Except otherwise noted, experiments all started from a pre-trained model, trained over 100 rounds. Each trial simulated 100 devices, with 10 percent of devices participating in each round and a single attacking device. While this is orders of magnitude smaller than a real FL deployment, it provides a good "worst-case" scenario, where the attacker controls 10% of the weight updates in a round. In the real world, an attacker may control many client devices. Each device received 20% of the dataset for training its local epochs.[3]

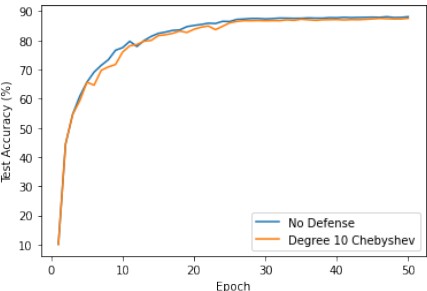

Figure 4: Model Training: using Chebyshev polynomial approximation vs. Unprotected.

**Evaluation of the noise attack.** Figure 5 (left) compares the efficacy of the Chebyshev aggregation schemes with the "ideal" sigmoid $H$ scheme and the simple averaging against the noise attack, when executed against a single round of FL.[4] The noise is evenly distributed on the range $[0, 1]$, so rejecting such an update should be trivial, as at least some weights are effectively guaranteed to fall outside the thresholds, which on some layers were set as small as $[0.1, 0.1]$. An attacker might try to get away with using smaller updates, but unless the attacker controls a huge percentage of devices,

---

[3]The full implementation of our Chebyshev and polynomial approximatons, FL simulation, and FHE computations, along with the code used to generate all of the graphs, is available on Github.

[4]The red part in the figures indicates the number of epochs during which the attacker is active, and so the attack shown to the left of Figure 5 only lasts 1 epoch, given that we are interested in seeing how much an attacker can degrade the test accuracy within a single epoch. When prolonging the attack throughout more epochs, the test accuracy without defense is not able to recover.

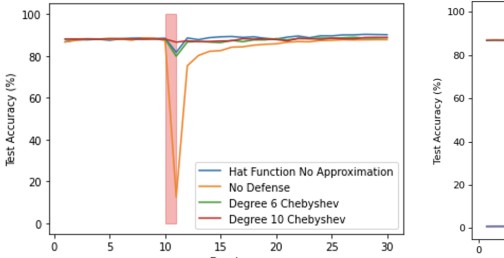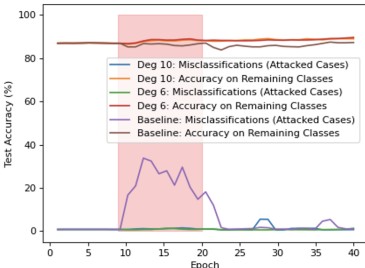

Figure 5: Defending against the noise attack (left) and defending against the switch classes attack (right).

these updates will have negligible effect on the global model. Note that the degree 10 Chebyshev approximation appears to perform better than the "ideal" $H$ function. This is merely a consequence of the fact that by the nature of the approximation, the range of updates allowed by the Chebyshev approximation is in fact slightly narrower than those allowed by the ideal $H$ (see Figure 2). The first concern that arises upon seeing this graph is that the 10 degree approximation is too aggressive— namely, that even reasonable updates might be rejected. However, this is not the case: Figure 4 compares the model accuracy when trained from scratch over 50 rounds of the undefended system and the degree-10 Chebyshev scheme. Observe that the model trains effectively as well, suggesting that we in fact could have set the allowable bounds of our simulation even narrower than we did.

**Evaluation of the class-switching attack.** Here, we are interested in two different metrics to evaluate the success of the attack: the accuracy of the model on the non-attacked classes and the percentage of elements of the attacked classes that are misclassified. Figure 5 (right) shows both of these metrics on the same axes, comparing the undefended and Chebyshev-defended models. Across both of these attacks, both the degree 6 and 10 Chebyshev approximations performed at least as well as the exact $H$ function, suggesting that low degree polynomial approximations are sufficient. While these experiments are promising, it is important to note their limitations. The CIFAR-10 dataset contains only 10 classes, so the model is very robust to small perturbations in weight values. It may be the case that on larger and more complex models, the aggregation error introduced by the weighting functions might have a larger effect on model accuracy than it does here.

## 6 CONCLUSION

The approach presented in this paper successfully defends against backdoor attacks, and—at least for the CIFAR 10 model on which we experimented—degree 6 polynomial approximations are sufficient. Furthermore, we show that they can be evaluated accurately and quickly with FHE. Still, we acknowledge the limitations of our method SAFHE, such as the inefficiency of FHE in some settings and the fact that the encryption space and the hat function $H$ space need to be close in order for the Chebyshev and minimax approximations to work correctly. For example, FHE schemes like CKKS can achieve speed ups by allowing SIMD by packing multiple plaintext elements in one cipher-text, and we could perhaps leverage such a speed-up inside of SAFHE.

Moreover, while we believe that our empirical evaluation demonstrates the feasibility of SAFHE (and applying the function $H$ is theoretically equivalent to the gradient norm clipping defense, which has been extensively tested in the literature, and so re-testing this fact is not part of our evaluation goals), testing SAFHE on other architectures and datasets would be beneficial, especially to see how the appropriate polynomial degree varies in each situation. Likewise, it would also be valuable to test SAFHE on other attacks, although we re-emphasize that our method is specifically engineered around the $\ell_2$ gradient norm attack, which is why we only test against the attack of malicious clients sending ill-formed gradients. It would be especially interesting to investigate the power of adaptive adversaries, although recent work has shown that the gradient clipping norm defense is effective against adaptive attacks Shejwalkar et al. (2022). Finally, choosing the $[a, b]$ parameters and the encryption parameters appropriately can be challenging in practice, which can also impact the effectiveness of our SAFHE method in some scenarios where they cannot be determined empirically.

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
