# OpenReview forum: "SAFHE: Defending Against Backdoor and Gradient Inversion Attacks in Federated Learning"
_ICLR.cc/2024/Conference — ICLR 2024 Conference Withdrawn Submission_

### Official Review · Reviewer_wYb4 · 2023-10-27

**Soundness:** 1 poor
**Presentation:** 2 fair
**Contribution:** 1 poor
**Rating:** 1
**Confidence:** 5

**Summary:**

The paper looks at simple defenses against poisoning in FL that rely on filtering out gradient updates with large norms. The authors propose to extend these defenses to the setting of secure aggregation with FHE by proposing a continuous polynomial approximation of the filtering-out operation called SAFHE, allowing a combined defense against both gradient leakage and poisoning. The authors show that their FHE approximation to filtering is accurate when degree 10 Chebyshev polynomials are used and that the aggregation scheme can work in theory in typical FL setups.

**Strengths:**

- The paper proposes a method that makes sense and describes it descently.

**Weaknesses:**

- **There are little to no contributions**
The method of applying well-known polynomial approximations such as the Taylor series in order to evaluate non-polynomial continuous functions with FHE is standard and even proposed in the original CKKS paper [1]. The authors use Chebyshev polynomials instead, but I would not call this contribution, as it is a straightforward change, which the authors do not properly evaluate (no experimental results from other schemes are available). The approximation of the clip function with the sum of sigmoids is also fairly standard (after all, sigmoid was proposed as a continuous version of the step function, and the clip function is just a composition of 2 step functions). Further, using FHE for aggregating updates in federated learning is standard, and the $\ell_2$-defense is very simplistic and very well known.
- **The evaluation is too simple**
The authors do not even implement the proposed defense end-to-end and instead implement their FHE filtering outside of federated learning. As such, their paper misses important details of how one would instantiate SAFHE in practice, such as the fact that $H$ needs to be applied on the norm of $w_i$ (at least if the defense is based on $l_2$ gradient norm and not the size of individual entries of the gradient which is how the authors present it) and not $w_i$ itself. The difference between the norm and $w_i$ itself is significant, as the norm computation requires additional multiplication operations per gradient, which can seriously affect the modulus degree required for good gradient estimation and, thus, the overall runtime. Further, the authors only evaluate the FHE computation on random numbers in $[-5,5]$, which is not anywhere close to how gradient entries or norms during training look like, and claim $10^{-6}$ FHE precision, which might not be enough in real deployment in the context of gradients of real networks.
A further way in which the evaluation is too simplistic is that it doesn't compare to exact norm filtering and doesn't report the exact values of $c$ used. Further, the only ablation provided is with respect to the polynomial degree with only two values (6 and 10) despite the fact that FHE experiments on the approximation of $H$ are super fast. Different polynomial approximations like the MinMax suggested throughout the text, and the effect of different values for $a,b$, and $c$ are not experimented with.
Finally, the federated learning experiments are too simple too. The authors use a single dataset without heterogeneity, which they attack with very simple attacks for only a single round. Further, the authors call the evaluated attacks backdoors, but they really represent simple data/model poisoning. The authors should switch to the poisoning language as backdoors assume that the overall accuracy of the model is unchanged and only the behavior on a handful of samples containing a trigger is.
- **Shortcomings of the method**
The proposed method assumes a known range of the norms of benign gradients, which they suggest estimating from unencrypted gradients of benign clients. Having unencrypted gradients and knowing which gradients are benign and which are not are strong assumptions that really defeat the purpose of applying the proposed defense in the first place.

**Questions:**

- What values of $c$ are you using in your experiments? Can you provide ablation with different values of c?
- Can you provide precision and time for evaluating the FHE **full computation** of $H$ - that is compute the $l_2$-norm and its weight of a real gradient from your network?
- Can you provide experiments with all polynomial degrees between 2 and 10? The experiments are embarrassingly fast so there is no reason not to.
- Can you do the above experiment for multiple values of the polynomial degree?
- Can you update the algorithm such that it estimates $a,b,c,d$ on encrypted gradients and without knowledge of which are benign and which are not?
- Run your experiments on more datasets and more models.
- If you don't want to evaluate the end-to-end FHE setup, can you at least account for the noise from FHE computations due to truncation by adding a uniform noise of that size to your gradients in the experiments of Figure 4 and 5?

All in all, this paper doesn't meet the criteria for acceptance in ICLR. The proposed method is, in my opinion, obvious and presents little real scientific contributions. The experiments are simple and do not even fully cover the complexity of applying the proposed defense to a realistic setup, like computing the norms of the gradients before applying $H$. I would have given the paper an even lower grade, but 2 is not allowed this year, and 1 would have been harsh, as the paper's proposed method at least makes sense. Note that answering the questions above is unlikely to change my grade. However, those questions need to be answered for this paper to be accepted in any venue like a workshop, a journal, or a lower-tier conference.

[1] Cheon, Jung Hee, et al. "Homomorphic encryption for arithmetic of approximate numbers." Advances in Cryptology–ASIACRYPT 2017: 23rd International Conference on the Theory and Applications of Cryptology and Information Security, Hong Kong, China, December 3-7, 2017, Proceedings, Part I 23. Springer International Publishing, 2017.

---

> ### Author Response · Authors · 2023-11-23
> **Part I**
>
> We thank the reviewer for the thorough comments on our paper, which will greatly help us improve our work.
>
> **First weakness bullet point**
>
> We agree that using Taylor approximations in order to evaluate non-polynomial functions is not a new idea in FHE, and it makes a lot of sense that it isn’t (and that similar techniques were used in the original CKKS paper), given that for FHE to be efficient in practice we can essentially only perform additions and multiplications. (This is the reason why we need to approximate the “hat function” H as a polynomial in the first place rather than using the product of two sigmoids directly.) We are not claiming that this alone is a contribution of our paper. As the reviewer mentions, we use Chebyshev polynomials instead; this is not because we are trying to make a more “creative” approximation than a Taylor approximation. Rather, the reason is that in the case of Taylor approximations, one can only guarantee closeness to the original function inside of the interval of convergence (which cannot be modified), whereas in our case we want to be able to approximate the function within an *arbitrary* interval [a, b]. (We mention this in Section 2.3.) This is why there is no experimental comparison between Taylor and Chebyshev approximations, and we see no reason for why we would evaluated this. The reason why we use Chebyshev polynomials rather than minimax polynomials in our experiments is because Chebyshev polynomials minimize the total approximation error, whereas Minimax polynomials minimize the maximum approximation error. Thus, Chebyshev polynomials are more suited for rejecting gradient updates that are outside of the [a, b]. This is again for a purely theoretical reason, which is why there is no need to compare these approximations empirically.
>
> Likewise, we do not claim that we are the first to think of expressing a step function H as a product of two sigmoids; that is the simplest way that we came up with to express the “hat function” as a polynomial (which we needed to do for the purposes of FHE efficiency). We agree that using FHE in FL settings is also a well-known technique, and that the $\ell_2$ norm defense is also well-known. But this is precisely why we think that SAFHE is a very valuable method: both FHE and the $\ell_2$ gradient norm are well-known defenses that have been extensively studied; FHE prevents gradient inversion attacks and $\ell_2$ defends against backdoor attacks. However, there is no method in the literature that proposes a way to combine these two defenses, and that is what we claim is our main research contribution. (Moreover, in general, any new method will inevitably combine building blocks that already exist, and we do not see why that would be regarded as a weakness of the method.) If the reviewer is aware of a paper that also proposes a combination of these two methods, then we would appreciate a pointer to the paper, because we are not aware of any such work. This is why we think that SAFHE is an interesting method: a priori, it seems impossible that one would be able to apply the $\ell_2$ gradient norm defense to encrypted gradients, precisely because they are encrypted. Moreover, as we emphasize in Section 6.2, while SAFHE focuses on the $\ell_2$ gradient norm defense, the key idea behind SAFHE is more general: we can adapt existing backdoor defenses from the literature to work with encrypted gradients if we are able to express the defense as a combination of polynomial evaluations, given that in an FHE setting, the server cannot *see* the gradients, but it can apply functions to the gradients.
>
> Moreover, we disagree with the claim that the $\ell_2$ gradient norm defense is very simplistic. The recent literature on backdoor attacks has deemed it the most effective attack in practice, and hence the one that is most important to protect against. This has recently been studied extensively in, for example, the papers “ACORN: Input Validation for Secure Aggregation” (USENIX’23), “RoFL: Robustness of Secure FL” (IEEE S&P’23), and “ELSA: Secure Aggregation for FL with Malicious Actors”, which clearly state that “filtering gradients based on their $\ell_2$ norm is effective against a large number of sophisticated poisoning attacks under realistic threat models for production FL” (ELSA, page 1). This is why both RoFL and ELSA (two of the most recent papers in this literature) focus on $\ell_2$ norm bounding as their defense, and this is the only attack that they consider.

---

> ### Author Response · Authors · 2023-11-23
> **Part II**
>
> Similarly, the RoFL paper states that “Prior work has identified enforcing norm bounds on individual client updates as a computationally simple yet promising defense in practical FL setups. This makes them a prime candidate [...]. Norm bounds, while not a panacea, have already been shown to prevent untargeted poisoning attacks in real-world adversarial scenarios.” After performing an extensive study of FL backdoor security, they conclude that “while they have clear limits, norm bounds would indeed be an attractive robustness solution” (RoFL paper, page 2), and they show how norm bounds are effective in preventing a class of highly practical attacks. Many other papers make this claim, such as “Back to the drawing board: A critical evaluation of poisoning attacks on FL” and “Can you really backdoor FL?”. So, while no known defense is enough to stop all poisoning attacks, the fact that the model poisoning literature has claimed this defense to be the most successful one in practice (including in the case of adaptive and untargeted attacks) is the reason why we engineered SAFHE around it. Even in the case of untargeted attacks, recent papers have shown “the high effectiveness of norm bounding against untargeted attacks in practice” (RoFL paper, page 3), and the $\ell_2$ norm bound defense “can prevent single-shot attacks for all adaptive attack strategies that we study” (RoFL, page 4).
>
> **Second weakness bullet point**
>
> Regarding the second bullet point on the simplicity of the evaluation, we agree that an end-to-end evaluation would have been preferable; however, our goal was to make a conceptual contribution (and to make the case that backdoor defenses can indeed be combined with encrypted gradients) and to show that SAFHE could be instantiated in practice. Given that our “hat function” corresponds *exactly* to the $\ell_2$ gradient norm defense, we defer to the extensive empirical evaluation that has been performed in the backdoor attack literature (e.g., ACORN, RoFL), as the effectiveness of the $\ell_2$ defense in real-world practical settings carries over to the SAFHE method. What matters in our setting is finding the right trade-off for determining the degree of the polynomial approximation of the hat function (the higher it is, the better the accuracy, but the slower the FHE). This is why we considered it sufficient to test only one dataset. That said, we agree that more evaluation is always preferable. We thank the reviewer for the wonderful suggestions on the different things that we should be testing in future work. Regarding the fact that H is actually applied to the norm of $w_i$ and not $w_i$ itself, we want to emphasize that in our experiments we are applying H to the norm of the gradients, so there shouldn’t be a worry about a mismatch here. As we state in the beginning of Section 2.1, while we write $w_i \in \mathbb{R}$ for simplicity in the presentation, our H function is applied to the $\ell_2$ norm of the client’s vector of weights. Similarly, our FHE experiment is applied to vectors of size 8192. We do not understand (and disagree) with the reviewer’s claim that evaluating the FHE computation on random numbers in [-5, 5] is “not anywhere close to how gradient entries or norms during training look like”. We chose the values [-5, 5] precisely because the typical gradient norms that we were seeing in our setting were roughly contained within this interval. The fact that the numbers are *random* in this interval in our FHE evaluation does not matter for how fast FHE is performing, given that this randomness does not affect the magnitude of the gradients (this is a small interval, so all of the random gradients are close).
>
> We agree that it would be interesting to compare SAFHE to the exact norm filtering; the reason why we didn’t do so is because SAFHE was already highly successful in preventing the attacks (and the exact norm filtering would only be better, given that the only difference between the two is in the Chebyshev approximation of H). Moreover, we emphasize that this is again a question of finding the right trade-off between the degree of the Chebyshev polynomial and the efficiency of FHE, and so this is 1) highly dependent on each particular setting, and 2) it is a *design choice*: in a particular setting, would one want to favor a faster end-to-end deployment at the expense of SAFHE being less accurate than the usual $\ell_2$ norm defense? (Given that a lower Chebyshev polynomial degree implies less accuracy.) Thus, there is no “right answer” on how to implement this trade-off in practice. The value of c that we used in all of our experiments is 0.005 (as it can be seen in our code), but we do not see why the reviewer is concerned about this: the value of c simply corresponds to the “steepness” of the hat function, and so one should just set it to be as small as possible – there is nothing else of interest to evaluate empirically here.

---

> ### Author Response · Authors · 2023-11-23
> **Part III**
>
> Regarding the polynomial degree, in our particular setting we found that degrees 6 to 10 provided great accuracy and efficiency, which is why that is the polynomial degrees for which we reported our experiments. We explained above why we report our experiments only with Chebyshev polynomials and not minimax polynomials – on a mathematical level, they quantify the error of the approximation differently, and the Chebyshev approximation is more suitable in the “hat function” approximation setting. Lastly, the reason why we don’t experiment why different [a, b] values is because we consider that to be an issue related to the original $\ell_2$ gradient norm defense, and not one related to SAFHE specifically. Several papers from the backdoor attacks literature contend with this problem, such as the paper “Robustness of Secure Federated Learning” by Lycklama et al. (IEEE S&P’23). We remark that other papers that study the problem of making the $\ell_2$ gradient norm defense secure against gradient inversion attacks also defer the problem of choosing the right [a, b] parameters to the backdoor attacks literature; e.g., the ELSA paper states that  selecting an appropriate gradient bound is “orthogonal” to the problem of making the method compatible with gradient inversion defenses. Therefore, when choosing the right [a, b] parameters, one should follow the suggestions already made in the $\ell_2$ gradient norm literature; these same recommendations apply to the case of SAFHE.
>
> We appreciate and agree with the reviewer’s distinction between backdoor attacks and model poisoning, and we will be more specific about this disctinction in the paper.
>
> **Third weakness bullet point**
>
> Again, the problem of estimating the [a, b] range is an issue of the $\ell_2$ gradient norm defense, not of SAFHE. Of course, in practice the server won’t have access to the unencrypted gradients in order to be able to make this choice. As explained in the previous paragraph, there are several papers in the backdoor attacks literature that study precisely this problem of estimating the range of the norms of benign gradients, which is why we do not re-study them in the case of SAFHE and consider it to be an orthogonal question.
>
> **Questions**
>
> The value of c is 0.005. As explained above, c is the steepness if H and so it should be as small as possible – no more empirical considerations are needed. We do not see the point in providing the evaluation numbers for all of the polynomial degrees, given that the empirical evaluation is meant to demonstrate that there exists *some* polynomial degree that provides a good trade-off between accuracy of the $\ell_2$ gradient norm defense and efficiency of the FHE evaluation. In our setting, this corresponds to degree 10. The question on why we do not investigate how to estimate [a, b] from the encrypted gradients in our experiments is answered in the previous paragraph. The rest of questions provide us with excellent suggestions for how to improve the empirical evaluation of our SAFHE method, and we will make sure to investigate those in the next iteration of our paper. We again thank the reviewer for these suggestions.

---

> ### Comment · Reviewer_wYb4 · 2023-11-23
>
> **First weakness bullet point:** The authors acknowledge that they do not have **ANY** technical contribution beyond just combining 2 very simple and well-known prior techniques in the most straightforward way possible. As the rest of my review points out, this is not even done in a particularly compelling way in terms of experiments and discussions. This alone makes this paper awfully inadequate for a top-tier ML conference. I am willing to argue with both the other reviewers and the ACs for the papers' rejection.
>
> The authors say their technique is general and can be applied to other backdoor defenses. This is patently false, as the only technical contribution they have is the polynomially approximating their proposed step function, which needs to be done from scratch for another defense, and there are multiple defenses with which it would not even be applicable as a strategy - e.g. KRUM [1].
>
> The authors say their norm defense is not simplistic. That is patently wrong. It is the most simple defense in the space. There is a very large body on different defenses in this space that are principled and allow much better protection like [1-5] and many more. It is true that SecureAggregation, in particular, struggles to integrate these more interesting defenses into their model. However, I think FHE should be able to integrate much more than simple l2-norm filtering.
>
> The authors say that their polynomial approximation is theoretically better, which means "there is no need for experiments". This is absolutely wrong. There is a need to determine how much better the proposed polynomial expansion is. This is especially true as this is one of the few things the authors really propose in this paper, since, by the authors own admission, this experiment takes seconds to execute, and since they themselves bring different schemes and claim practically they observed they use the best one. Quite frankly, the reviewer feels insulted by this answer. Further, the authors should also discuss which polynomial approximation schemes have been used in the vast literature of FHE, when and why.
>
> **Second weakness bullet point:** I disagree with the authors that they make conceptual contributions. The idea of combining l2 norm filtering with FHE is trivial. Further, for proper contribution, they need to properly test their idea which the authors do a poor job at. My range argument is that most gradient entries are much smaller than 5 in absolute value - more like $0.1$ and below. This is the range I want to know the authors' method works at.
>
> **Third weakness bullet point:** This is absolutely a problem of this paper. In no FHE is used in this setup, the range is easy to predict, as the authors admit. If they find that there are works that can be combined with FHE to predict the range, they should show this FHE instantiation. This is, after all, part of the core and sole contribution of this paper - making l2 norm filtering possible with FHE. The reviewer finds this answer weak at best.
>
> I have decreased my score to 1. I am not satisfied with the answers of the authors, and I was considering 1 before, given the lack of a score of 2 at ICLR anyways. The authors evaded all of the suggested experiments. The reviewer is now convinced that the results of those suggested experiments will not paint the paper in a good light.
>
> [1] Peva Blanchard, El Mahdi El Mhamdi, Rachid Guerraoui, and Julien Stainer. Machine learning with adversaries: Byzantine tolerant gradient descent. In NIPS, 2017.
> [2] Dong Yin, Yudong Chen, Ramchandran Kannan, and Peter Bartlett. Byzantine-robust distributed learning: Towards optimal statistical rates. In International Conference on Machine Learning, pages 5650–5659. PMLR, 2018.
> [3] Banghua Zhu, Jiantao Jiao, and Jacob Steinhardt. Robust estimation via generalized quasi-gradients. Information and Inference: A Journal of the IMA, August, 2021.
> [4] Banghua Zhu, Jiantao Jiao, and Michael I Jordan. Robust estimation for nonparametric families via generative adversarial networks. arXiv preprint arXiv:2202.01269,2022.
> [5] Raj Kiriti Velicheti, Derek Xia, and Oluwasanmi Koyejo. Secure byzantine-robust distributed learning via clustering. arXiv preprint arXiv:2110.02940, 2021.

---

### Official Review · Reviewer_cV7D · 2023-11-05

**Soundness:** 3 good
**Presentation:** 3 good
**Contribution:** 3 good
**Rating:** 6
**Confidence:** 2

**Summary:**

This paper present SAFHE (secure aggregation with fully homomorphic encryption), which can defend against gradient inversion attacks since plain-text updates or decryption keys cannot be seen; and it can defend against backdoor attacks by rejecting large weight updates in FHE domain without conditional branches. The work proposes an approach to find an appropriate weighting function, by approximations of Chebyshev and Minimax, and prove it to be both effective and efficient.

**Strengths:**

1. The paper introduces a clear and novel concept by proposing a weighting function that determines whether to accept (1) or reject (0) client updates, with the ability to compute it in the FHE setting. The authors have made significant progress in finding effective approximations for this weighting function. However, I must note that my familiarity with FHE is limited, which may affect my ability to fully grasp the context and accurately assess the paper's contribution.

2. Writing is good and easy to follow.

**Weaknesses:**

1. Mistakes in the paper: Figure 2 has "mistakes to fix" that H function does not appear similar in left and right subfigures.

2. "Gradient inversion attacks" appear in the title, but the whole body of the paper mainly deals with backdoor attacks (how to reject too large gradients), and does not elaborate on how it can defend against gradient inversion attacks.

**Questions:**

1. I have limited knowledge on FHE, so I am not sure about "gradient inversion attacks can not happen if not providing plain-text gradients".  I believe up to now most gradient inversion attacks happen in plain-text gradients, but does that say gradient inversion will be impossible if using FHE? Is it theoretically guaranteed? Can the authors refer some more materials for me to understand why it is the case?

2.  The success of SAFHE relies on the assumption that gradients updates of a backdoor attack are large; what if the adversary optimize their decoy gradients to be small? And how about benign but out-of-distribution samples / hard samples, which could introduce large but benign gradient updates?

I think the paper is interesting. I am willing to raise my scores if my concerns and questions are solved by authors.

---

> ### Author Response · Authors · 2023-11-23
> **Part I**
>
> We thank the reviewer for their insightful comments, which have already helped us improve our paper!
>
> **Regarding the weaknesses,** we thank the reviewer for catching our note regarding Figure 2. As for the comment on the weight of the gradient inversion attacks in the paper, we agree that most of the body of the paper is devoted to the $\ell_2$ gradient norm defense and backdoor attacks. The reason is that, as we elaborate in the paragraph below, using FHE as part of our method directly prevents gradient inversion attacks, given that all of the weights that are sent from the client to the server are *encrypted*, and gradient inversion attacks try to reconstruct the clients’ private data from the *plaintext* (i.e., unencrypted) gradients. Therefore, these attacks cannot succeed without having access to the unencrypted gradients. We do not need to modify any existing FHE scheme for our purposes, which is why we do not need to say more about gradient inversion attacks. What is complicated and interesting is how to combine encrypted gradients with current existing backdoor defenses (such as the $\ell_2$ gradient norm defense, which is our focus in the paper), given that a priori these require examining the *plaintext* gradients. This is why we wanted to say both “gradient inversion” and “backdoor attacks” in the title, as to emphasize that our research question investigates how to defend against both of these types of attacks *simultaneously*. However, we will make sure to state more clearly how SAFHE defends against gradient inversion attacks by virtue of FHE, and we agree with the reviewer’s concern that this is not clear enough in the body of the paper. We should also provide a more thorough explanation on FHE.
>
> **Regarding the question on FHE,** fully homomorphic encryption is a cryptographic technique that allows us to encrypt data while still being able to perform computations on the *encrypted* data, without ever having to decrypt it in the process. We can see FHE as an extension of public-key cryptography: a party first encrypts the data with their private key, and then another party is able to perform computations on the data *without* having access to the secret key. In our setting, the first party corresponds to each of the clients (each client has their own private key), whereas the latter party corresponds to the server. FHE is thus an extremely powerful cryptographic technique (and in fact seemed impossible to achieve until Craig Gentry surprised the cryptographic community with his paper “Fully Homomorphic Encryption Using Ideal Lattices” in 2009). What is important to emphasize is that FHE is an *encryption scheme*, and thus in our SAFHE method all of the gradients that the clients send to the server an encrypted. On a theoretical level, by the mathematical guarantees of any secure encryption scheme, the encrypted gradients cannot be decrypted by a party that does not possess the secret key (which applies to any malicious eavesdropper that is trying to perform a gradient inversion attack). As described in the seminar paper “Deep Leakage from Gradients” (Zhu et al. NeurIPS’19) and its follows-up, a gradient inversion attack consists of an eavesdropper attempting to reconstruct the client’s private data from the gradient (given that the gradient has been computed using the client’s private data, and thus is encoding private information). See also, for example, the paper “Evaluating Gradient Inversion Attacks and Defenses in Federated Learning” (Huang et al. NeurIPS’21), which illustrates how all of the gradient inversion attacks follow this recipe. Therefore, yes, gradient inversion attacks are theoretically guaranteed to be impossible to carry out if the gradients have been encrypted with a secure encryption scheme (as it is the case of FHE), and thus in a setting where the eavesdropper cannot access the unencrypted gradients. (Indeed, per the security of the encryption scheme, the gradients look like "random strings" to the eavesdropper.) The reason why preventing gradient inversion attacks is still an active area of research is because FHE, while provably secure as a cryptographic primitive, is usually inefficient in practice, especially for large scale deployments. However, alternative more efficient defenses which do not employ any cryptographic primitives are unlikely to provide any theoretical guarantees (e.g., none of the defenses considered in Huang et al. do). Moreover, in recent years, much more efficient and scalable FHE schemes have been proposed specifically tailored for FHE (see, e.g., the paper “FedML-HE” by Jin et al. 2023 and “BatchCrypt”by Zhang et al. USENIX’20), and we are hopeful that the efficiency of FHE schemes in practice will continue to improve. For example, NVIDIA and IBM’s FL systems already incorporate homomorphic encryption (https://arxiv.org/pdf/2210.13291.pdf).

---

> ### Author Response · Authors · 2023-11-23
> **Part II**
>
> **Regarding the second question,** we want to emphasize that SAFHE is engineered around the $\ell_2$ gradient norm defense from the backdoor attack literature. Indeed, our “hat function” H corresponds exactly to the $\ell_2$ gradient norm defense, but it is expressed as a polynomial evaluation instead. In the case of the $\ell_2$ gradient norm defense, unlike the case of FHE and gradient inversion attacks, there is no theoretical guarantee that it works against all malicious adversaries. However, this is not a bug of SAFHE, and is true for any backdoor defense in the literature. Indeed, the question of whether FL systems can be made robust against backdoors is currently a major open question in privacy-preserving research in FL (see, for example, “Attack of the Tails: Yes, You Really Can Backdoor Federated Learning” by Wang et al. in NeurIPS’20). As the reviewer points out, there can exist malicious adversaries that manage to perform a backdoor attack using only small gradients. However, as we summarize in the paper, the recent literature on backdoor attacks has deemed the $\ell_2$ gradient norm defense as the most effective backdoor defense in practice. This has recently been studied extensively in, for example, the papers “ACORN: Input Validation for Secure Aggregation” (USENIX’23), “RoFL: Robustness of Secure FL” (IEEE S&P’23), and “ELSA: Secure Aggregation for FL with Malicious Actors”, which clearly state that “filtering gradients based on their $\ell_2$ norm is effective against a large number of sophisticated poisoning attacks under realistic threat models for production FL” (ELSA, page 1). This is why both RoFL and ELSA (two of the most recent papers in this literature) focus on $\ell_2$ norm bounding as their defense, and this is the only attack that they consider. Similarly, the RoFL paper states that “Prior work has identified enforcing norm bounds on individual client updates as a computationally simple yet promising defense in practical FL setups. This makes them a prime candidate [...]. Norm bounds, while not a panacea, have already been shown to prevent untargeted poisoning attacks in real-world adversarial scenarios.” After performing an extensive study of FL backdoor security, they conclude that “while they have clear limits, norm bounds would indeed be an attractive robustness solution” (RoFL paper, page 2), and they show how norm bounds are effective in preventing a class of highly practical attacks. For example, even in the case of untargeted attacks, recent papers have shown “the high effectiveness of norm bounding against untargeted attacks in practice” (RoFL paper, page 3, citing the paper “Back to the Drawing Board: A Critical Evaluation of Poisoning Attacks on Production Federated Learning” from IEEE S&P’22). Likewise, the RoFL paper states that the $\ell_2$ norm bound defense “can prevent single-shot attacks for all adaptive attack strategies that we study” (RoFL, page 4).
>
> Lastly, as the reviewer points out, some benign large gradients might be rejected by SAFHE. However, we again want to emphasize that this is a problem with the $\ell_2$ gradient norm defense, and not a bug of SAFHE, which is concerned with making the $\ell_2$ gradient norm defense compatible with gradient inversion defenses. For the reason pointed out by the reviewer, it is very important to pick the “hat function” parameters [a, b] appropriately, so that benign gradients are not rejected. Several papers from the backdoor attacks literature contend with this problem, such as the paper “Robustness of Secure Federated Learning” by Lycklama et al. (IEEE S&P’23). We remark that other papers that study the problem of making the $\ell_2$ gradient norm defense secure against gradient inversion attacks also defer the problem of choosing the right [a, b] parameters to the backdoor attacks literature; e.g., the ELSA paper states that  selecting an appropriate gradient bound is “orthogonal” to the problem of making the method compatible with gradient inversion defenses.

---

> ### Comment · Reviewer_cV7D · 2023-11-23
> **Thanks for the response**
>
> Thanks authors for their rebuttal and clarifications, and their efforts to bring many interesting related works to my sight. Their answers partly solve my concerns and questions, so I am willing to raise my score towards above borderline. However, considering my missing knowledge in the context of FHE, I only have a low confidence in my judgement of technical contributions of this work, therefore the confidence score remains to be 2.

---

### Official Review · Reviewer_uyJ6 · 2023-11-07

**Soundness:** 2 fair
**Presentation:** 3 good
**Contribution:** 2 fair
**Rating:** 5
**Confidence:** 5

**Summary:**

This work proposed SAFHE, Secure Aggregation with Fully Homomorphic Encryption, a novel scheme to defend against both backdoor attacks and gradient inversion attacks. Their secure aggregation method combines the use of fully homomorphic encryption (FHE) and the gradient norm clipping defense to defend against large malicious client updates.

**Strengths:**

- The area and overall idea are interesting.
- The paper is well written, and the ideas are easy to follow for readers.
- The proposed scheme is interesting.
- Provide a detailed simulation study and provide a detailed benchmarking.

**Weaknesses:**

Lack of theoretical support for the security guarantees of the proposed scheme. There are no security proofs (or any proof sketch) and privacy guarantees discussion of their proposed framework.

**Questions:**

Does the proposed scheme support malicious threat model?

---

> ### Author Response · Authors · 2023-11-22
>
> We thank the review for reading our paper and for their helpful feedback. Regarding the lack of theoretical support, we want to point out that our scheme is secure because it uses a secure FHE scheme, which we do not modify in any way. Given that we directly use the CKKS scheme (Cheon et al. ASIACRYPT’17, “Homomorphic encryption for arithmetic of approximate numbers”) as implemented by the Microsoft SEAL library (https://www.microsoft.com/en-us/research/project/microsoft-seal/), the security proofs of the CKKS paper that show that this is a secure encryption scheme directly apply to our setting, which is why we did not deem it necessary to include any security proofs (again, we emphasize that we do are not modifying the underlying FHE scheme in any way, and therefore the FHE security guarantees directly apply to SAFHE). Therefore, when we consider gradient inversion attacks, the security of the underlying FHE scheme implies that we have the theoretical guarantee that the clients’ weights cannot be decrypted, and thus SAFHE defends against any gradient inversion attack, including in the malicious threat model.
>
> On the other hand, when we consider backdoor attacks, the reason why we do not include security proofs is because we cannot theoretically guarantee that SAFHE will be always successful in preventing any backdoor attack. The reason is that our method is based on the $\ell_2$ norm gradient defense, which, according to the most recent literature on backdoor attacks in federated learning, is highly effective in practice, but, like any other backdoor defense method, is not provably secure. Its effectiveness in practice has recently been studied extensively in, for example, the papers “ACORN: Input Validation for Secure Aggregation” (USENIX’23), “RoFL: Robustness of Secure FL” (IEEE S&P’23), and “ELSA: Secure Aggregation for FL with Malicious Actors”, and “Back to the Drawing Board: A Critical Evaluation of Poisoning Attacks on Production Federated Learning” from IEEE S&P’22, which clearly state that “filtering gradients based on their $\ell_2$ norm is effective against a large number of sophisticated poisoning attacks under realistic threat models for production FL” (ELSA, page 1). This is why all of RoFL, ELSA, and ACORN focus on $\ell_2$ norm bounding as their defense, and this is the only attack that they consider. Similarly, the RoFL paper states that “Prior work has identified enforcing norm bounds on individual client updates as a computationally simple yet promising defense in practical FL setups. This makes them a prime candidate [...]. Norm bounds, while not a panacea, have already been shown to prevent untargeted poisoning attacks in real-world adversarial scenarios.” After performing an extensive study of FL backdoor security, they conclude that “while they have clear limits, norm bounds would indeed be an attractive robustness solution”, and they show how norm bounds are effective in preventing a class of highly practical attacks. In particular, the RoFL paper states that the $\ell_2$ norm bound defense “can prevent single-shot attacks for all adaptive attack strategies that we study” (RoFL, page 4).
>
> Therefore, we cannot provide theoretical guarantees on the success of SAFHE in protecting against backdoor attacks because the $\ell_2$ norm gradient defense is also not a provable defense method (as opposed to FHE against gradient inversion attacks). None of the papers in the literature on $\ell_2$ gradient norm defenses is able to provide a theoretical guarantee. However, as we summarized in the previous paragraph and as we summarize in the paper, this is a highly successful defense in practice (which is also why we engineered SAFHE around it in the first place). Beyond the $\ell_2$ norm defense, we want to emphasize that backdoor attacks are very complicated to deal with, and that we are not aware of *any* defense against backdoor attacks in the literature that provably defends against malicious adversaries. Instead, papers in the area usually consider how successful the attacks are in practice. The question of whether FL systems can be made robust against backdoors is currently a major question in privacy-preserving research in FL (see, for example, “Attack of the Tails: Yes, You Really Can Backdoor Federated Learning” by Wang et al. in NeurIPS’20), so we want to emphasize that the fact that we cannot provide a theoretical guarantee of SAFHE against malicious adversaries is what one should expect to get given the current research status around backdoor attacks in FL.
>
> Still, we believe that SAFHE is a valuable addition to the privacy-preserving literature because it constitutes a method that is 1) easy to implement, 2) provably secure against gradient inversion attacks (as guaranteed by the security of FHE schemes), and 3) effective in practice against backdoor attacks (as extensively shown in the recent literature that has studied the effectiveness of the $\ell_2$ gradient norm defense).

---

> > ### Comment · Reviewer_uyJ6 · 2023-12-01
> > **Reply to Author Response**
> >
> > Thank you for the author's rebuttal and clarifications. But, their response doesn't address my concerns (and so I'm not raising my score).
> >
> > - The authors mentioned that their proposed method (SAFHE) is based on the FHE (which they using without any changes). However, it's important to provide a solid justification for how SAFHE provides guarantees for Correctness and Privacy, how SAFHE handles the malicious server case, etc.  (The authors mentioned some paper names in their response to justify why they cannot provide theoretical guarantees, but I want to mention that all those papers provide solid theoretical guarantees of their proposed methods).